# Small Mammals in Forests of Romania: Habitat Type Use and Additive Diversity Partitioning

**Anamaria Lazăr [1], Ana Maria Benedek [2,]\* and Ioan Sîrbu [2]**

[1] Department for Engineering and Management in Food and Tourism, Faculty of Food and Tourism, Transilvania University of Brașov, 148 Castelului Street, 500036 Brașov, Romania; anamaria.gurzau@unitbv.ro

[2] Faculty of Sciences, Lucian Blaga University of Sibiu, 5-7 Rațiu Street, 550012 Sibiu, Romania; ioan.n.sirbu@ulbsibiu.ro

\* Correspondence: ana.benedek@ulbsibiu.ro; Tel.: +40-744-538-278

**Abstract:** Small mammals are key components of forest ecosystems, playing vital roles for numerous groups of forest organisms: they exert bottom-up and top-down regulatory effects on vertebrate and invertebrate populations, respectively; they are fungus- and seed-dispersers and bioturbators. Therefore, preserving or restoring the diversity of small mammal communities may help maintain the functions of these ecosystems. In Romania, a country with low-intensity forest management and a high percentage of natural forests compared to other European countries, an overview of forest small mammal diversity and habitat type use is lacking, and our study aimed to fill this gap. We also aimed to partition the total small mammal diversity of Romanian forests into the alpha (plot-level), beta, and delta (among forest types) diversities, as well as further partition beta diversity into its spatial (among plots) and temporal (among years) components. We surveyed small mammals by live trapping in eight types of forest across Romania. We found that small mammal abundance was significantly higher in lowland than in mountain forests, but species richness was similar, being associated with the diversity of tree canopy, with the highest values in mixed forests. In contrast, small mammal heterogeneity was related to overall habitat heterogeneity. As predicted, community composition was most distinct in poplar plantations, where forest specialists coexist with open habitat species. Most of the diversity was represented by alpha diversity. Because of strong fluctuations in population density of dominant rodents, the temporal component of beta heterogeneity was larger than the spatial component, but species richness also presented an important temporal turnover. Our results show the importance of the time dimension in the design of the surveys aiming at estimating the diversity of small mammal communities, both at the local and regional scales.

**Keywords:** alpha, beta, and delta diversities; rarefaction; rodents; shrews; multivariate ordination; community composition; niche width

## 1. Introduction

One of the main goals of biological conservation is the identification and preservation of sites, habitats, and landscapes that act as biodiversity hotspots, hosting a high level of floristical and faunistical richness. The classical concept of diversity—the variety of organisms in a community, now known as alpha diversity—was developed to encompass multiple spatial scales: gamma diversity—the landscape scale diversity, and epsilon diversity—the diversity of entire geographic regions [1]. The degree of change in species composition among communities and landscapes is a measure of diversity itself, defined as beta and delta diversity. Since its definition in 1972 by Whittaker [2], beta diversity has become an important tool for understanding the origin, functioning, and maintenance of biodiversity at local and regional levels [3]. The landscape-scale (gamma) diversity results from the combination of community-level (alpha) diversity and among-community (beta) diversity. Diversity partitioning complements existing models in conservation biology and may provide a unique approach to understanding species diversity across spatial scales [4].

From a methodological point of view, there are various ways and techniques to partition the gamma diversity into its alpha and beta components. In the additive model of diversity decomposition, the gamma diversity represents the sum of the alpha and beta diversity. The additive decomposition of diversity indices has its drawbacks [5,6] but also some advantages. The transformation of decomposed variation into diversity measures may be useful for exploring contribution of individual species to diversity components or relating diversity patterns to environmental variables [7].

The most diverse ecological group of mammals, terrestrial small mammals, are fundamental components of most terrestrial ecosystems, including forests. Their main role is related to their position in the food chains. Most small mammals, especially rodents, are the main prey of many vertebrate predators, having a bottom-up regulatory effect on predator populations [8]. Shrews and some rodents may also exercise top-down control on the distribution, abundance, and fluctuations of insect or other invertebrate populations [9]. Rodents are also involved in fungus dispersal [10] and because of their caching behaviour, they are important seed dispersers. They tend to hoard seeds in microsites where the emergence of seedlings would be enhanced and their survival increased because of lower densities of conspecific trees [11,12], contributing to forest persistence, expansion and regeneration. Many small mammals are burrowers; therefore, they also have a bioturbation role [13], with an important effect on soil fauna and primary production. From an economic perspective, some forest rodents may be regarded as pests, as seed predation and tree bark consumption can affect forests, especially young plantations [14].

In Romania, 58.9% of forests are montane, distributed along the Carpathian Mountains, which are heavily forested, and only 6.5% are situated in the plains, where massive deforestations took place for the expansion of agricultural land; the rest are forests in hilly areas [15]. Unlike in western and central Europe, where intensive forest management has resulted in even-aged monocultures, especially of spruce and other conifers, including allochthonous species, forest management in Romania has not been so intensive, and thus the natural composition of tree species has generally been preserved. As a result, forests with more than 99% autochthonous trees represent 96.6% of Romanian forests, while plantations of allochthonous species represent only 1.4% and are established mainly in the plains [15]. The Southern Carpathian Mountains, where most of our montane study sites were located (Figure 1), shelter most of the country's old-growth forests (1.4% of all Romanian forests), some of them being included in the survey.

Lists of species and community structure data of small mammals in forests across Romania are available in the literature (e.g., [16,17]) but there have not been any studies performed at a regional level to evaluate forest type use by small mammals and their diversity. Therefore, we aimed to evaluate habitat use by small mammals in the most common forest types in Romania. We hypothesised that because of their peculiar abiotic conditions and vegetation structure, poplar plantations would shelter the most distinctive small mammal communities. In addition, we aimed to evaluate their diversity, in terms of species richness and heterogeneity, and test their relationship with elevation, canopy diversity, and habitat heterogeneity; partition forest small mammal diversity into its alpha, beta, and delta components; and compare spatial and temporal species turnover. We hypothesised that because of the strong population fluctuations, especially in rodents, temporal beta diversity would be an important source of large-scale diversity in forest small mammals.

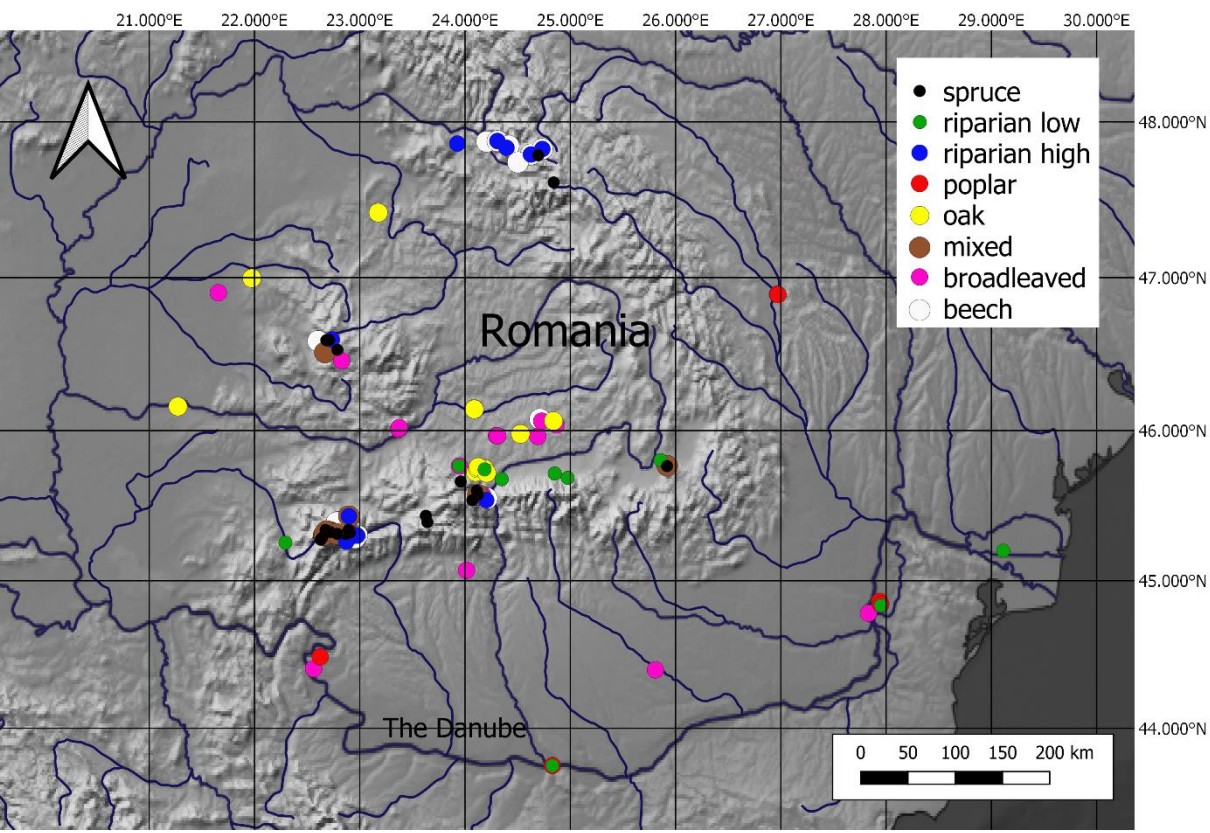

**Figure 1.** Location of sampled forests.

## 2. Materials and Methods

### 2.1. Habitat Description

The surveyed forest plots, distributed across Romania, were assigned to one of the following forest types: poplar plantations, oak forests, mixed broadleaved forests, lowland riparian forests, montane riparian forests, beech forests, mixed broadleaf and conifer forests, and spruce forests (Figure 1). We grouped these forest types into two categories, depending on elevation: the first four are lowland forests and the last four are montane forests. For each surveyed forest plot, we recorded the elevation as quantitative variable (expressed in m a.s.l.) and the diversity of the tree canopy and the overall habitat heterogeneity as ordinal variables. The canopy diversity was evaluated on the basis of the number and cover of species in the tree canopy as 1—only one species, 2—one dominant species (over 75% cover), 3—two dominant species (over 40% cover each), 4—three or more codominant species. The overall habitat heterogeneity was evaluated on the basis of the cover and composition of shrub canopy and herb layer, the abundance of coarse woody debris (CWD), and rocky outcrops and surfacing stones: 1—no understory, poor herb layer, no or little CWD and rocks; 2—sparse shrubs, abundant CWD or rocks; 3—closed and diverse understory, rich herb layer, abundant CWD or rocks or both.

### 2.2. Small Mammal Trapping

Small mammal trapping was conducted during the warm season (from June to October) between 2000 and 2018 across Romania, at elevations between 5 and 1710 m a.s.l., in the forest types mentioned above, described in Appendix A.

We live-trapped small mammals using artisanal wooden and plastic box-traps (18 × 10 × 8 cm). Transects included 30 to 40 traps set 15 m apart. Because many traps were disturbed by weather, animals, or people, the trapping effort differed greatly among transects, varying between 20 and 120 trap nights (TN) per transect. Traps were baited with sunflower seeds and apple slices, and insulated with hay. Traps were checked at dawn and

dusk for two or three consecutive days. We identified captured animals to species on the basis of morphological traits, marked them by fur clipping, and then released each at its trapping site. Recaptures were not considered in the analyses.

### 2.3. Data Analysis

The dataset used in this study came from different small mammal inventory programs, and therefore the design is not balanced. Thus, in most forest plots, traps were set only once, in different years, while some plots were randomly surveyed multiple times, with one mixed forest surveyed 60 times (for 12 years), resulting in very different trapping efforts allocated to the various forest plots and types. Therefore, to make data size comparable among forest types and to avoid spatial pseudoreplication, we selected randomly one transect from each forest plot that was repeatedly surveyed and we used only this dataset in the analyses, comprising between 6 and 24 transects per forest type, with a total of 110 transects (Table 1).

**Table 1.** Results of small mammal trapping in the selected transects of the eight surveyed forest types, with no temporal replicates.

| Forest Type | Lowlands | | | | Mountains | | | | |
|---|---|---|---|---|---|---|---|---|---|
| | Poplar | Oak | Broadleaved | Riparian | Riparian | Beech | Mixed | Spruce | Total |
| Transects | 6 | 12 | 15 | 11 | 11 | 12 | 19 | 24 | 110 |
| Empty transects | 1 | 1 | 1 | 1 | 1 | 3 | 4 | 7 | 19 |
| % Empty | 16.7 | 8.3 | 6.7 | 9.1 | 9.1 | 25.0 | 21.1 | 29.2 | 25 |
| *Apodemus agrarius* | 7 | 4 | 24 | 41 | 14 | 2 | 0 | 0 | 92 |
| *Apodemus flavicollis* | 11 | 118 | 125 | 25 | 75 | 28 | 33 | 74 | 489 |
| *Apodemus sylvaticus* | 2 | 2 | 9 | 0 | 0 | 2 | 0 | 0 | 15 |
| *Apodemus uralensis* | 1 | 0 | 0 | 0 | 0 | 0 | 0 | 0 | 1 |
| *Chionomys nivalis* | 0 | 0 | 0 | 0 | 0 | 0 | 3 | 1 | 4 |
| *Crocidura suaveolens* | 2 | 0 | 0 | 0 | 0 | 0 | 0 | 0 | 2 |
| *Glis glis* | 0 | 0 | 4 | 0 | 0 | 0 | 0 | 0 | 4 |
| *Microtus agrestis* | 0 | 0 | 1 | 0 | 1 | 0 | 0 | 1 | 3 |
| *Microtus arvalis* | 11 | 0 | 15 | 3 | 0 | 0 | 0 | 0 | 29 |
| *Muscardinus avellanarius* | 0 | 1 | 1 | 0 | 1 | 0 | 3 | 1 | 7 |
| *Myodes glareolus* | 0 | 3 | 15 | 3 | 15 | 20 | 57 | 49 | 162 |
| *Mus musculus* | 0 | 0 | 0 | 0 | 1 | 0 | 0 | 0 | 1 |
| *Microtus subterraneus* | 0 | 1 | 0 | 1 | 0 | 0 | 1 | 1 | 4 |
| *Neomys fodiens* | 0 | 0 | 1 | 1 | 0 | 0 | 0 | 0 | 2 |
| *Sorex alpinus* | 0 | 0 | 0 | 0 | 0 | 0 | 2 | 0 | 2 |
| *Sorex araneus* | 0 | 0 | 3 | 1 | 0 | 2 | 19 | 12 | 37 |
| *Sorex minutus* | 0 | 0 | 0 | 2 | 1 | 0 | 2 | 1 | 6 |
| Total individuals | 34 | 129 | 198 | 77 | 108 | 54 | 120 | 140 | 860 |
| Mean capture index | 13.5 | 25.3 | 25.6 | 21.9 | 34 | 12.2 | 13.2 | 14.7 | |
| Total species | 6 | 6 | 10 | 8 | 7 | 5 | 8 | 8 | 17 |
| Mean no. of species per transect | 2.16 | 1.5 | 2.4 | 2 | 1.72 | 1.25 | 1.79 | 1.45 | |
| Estimated richness (bootstrap estimator) | 8.5 | 8.75 | 13.73 | 11.64 | 10.63 | 5 | 8.95 | 12.79 | |

Because the number of captured individuals is strongly influenced by the trapping effort, we calculated the capture index as the number of captured individuals per 100 effective trap-nights and used it as a proxy for abundance. To calculate the effective trap-nights, we subtracted from the total number of trap-nights those that were non-functional or occupied by recaptured individuals. We tested the difference between lowland and montane forests in species richness (number of captured species), heterogeneity (Simpson index), abundance of common species (with more than 20 captured individuals), and total abundance using the non-parametric Wilcoxon test because the data normality assumption was

not met. The difference between lowlands and mountains in the empty transects was tested using the chi-squared test. To test the effect of habitat characteristics on small mammal diversity, we used generalised linear mixed models in package lme4 [18] in R version 3.6.1 [19], with species richness and heterogeneity as response variables, respectively; elevation, diversity of tree canopy, and habitat heterogeneity as predictors; and year as a random factor. We used Poisson distribution for species richness and binomial distribution for heterogeneity. The significance of the predictors was tested by the likelihood-ratio test (LR test).

We calculated the relative occurrence of small mammal species in the forest types as the proportion of the individuals of each species captured in the various forest types during the whole study period. To evaluate the niche width of the small mammals, we used the FT Smith index [20], calculated in R, with the forest types as resources, the relative occurrences as the proportions of individuals exploiting these resources, and the proportion of trapping effort used in the habitat types as the resource availability.

The response of small mammals to forest type at community level was analysed using Canoco 5.12 software (supplied by Microcomputer Power, Ithaca, New York, NY, USA) [21], including the number of captured individuals of each species per transect as response variable and the habitat type as predictor. An indirect gradient analysis, the detrended correspondence analysis (DCA) was first performed to establish the length of the gradients and to summarise the variation in the small mammal community. Because the length of the longest gradient was 6.3 standard deviation (SD) units, we used canonical correspondence analysis (CCA), which is suitable for the analysis of datasets with high species turnover. However, in CCA, response data are standardised by site total; therefore, the results refer to the relative abundances of species, which we refer to as species composition hereafter. Response data were log-transformed by the expression $y' = \log(y + 1)$. When reporting the effects of various forest types on species composition, we adjusted probabilities ($p_{adj}$) to correct for the inflation of type-I error caused by multiple testing, using the false discovery rate values [7]. The significance of ordination axes was tested by the Monte Carlo permutation test with 999 permutations per test. Because data were collected over several years and small mammal communities exhibit strong annual fluctuations, we used the variation partitioning procedure to assess and compare the explanatory importance of the forest types and year of survey, measuring and testing their unique (conditional) effects and evaluating their overlap.

Significance of response of individual species to the forest types was illustrated by means of t-value biplots, which approximate the t-values of the regression coefficients of a multiple regression with the particular species as response variable and all the habitat types as predictors, revealing statistically significant pair-wise relationships between each response variable and each predictor [7].

As measure of species richness, we used the number of captured species per transect, and as measure of community heterogeneity, we used the Simpson index of diversity [20]. To evaluate the efficiency of our sampling in species richness estimation, we constructed the rarefaction curves for each forest type using the function specaccum in vegan package [22]. Because of the differences in the sample sizes (different trapping efforts among transects), we used the individual-based approach for constructing the rarefaction curves.

The additive decomposition of diversity may be performed through ordination methods applied on the inflated data table (instead of the standard response data table describing community composition for each site), in which each row represents a single species occurrence, i.e., one non-zero cell of the original species data table, and the weights represent the number of individuals or any other measure of species abundance [7]. The total variation in the response data obtained using a weighted principal component analysis (PCA), divided by the sum of the weights, represents the estimate of total (gamma) diversity expressed as Simpson index [23]. For the estimation of beta diversity, a redundancy analysis (RDA) with site membership as explanatory variable must be performed, and the percentage of variation explained by the constrained axes represents the contribution of the beta

component to the total diversity [7]. Decomposition of species richness may be performed more easily using the unimodal ordination methods. The total variation reported by the correspondence analysis (CA) applied to the weights is equal to the gamma species richness minus one, and in the CCA with site membership as explanatory variable, the explained variation is the beta richness minus one [7].

To partition the diversity (species richness and community heterogeneity) of Romanian forest small mammals in its spatial and temporal components, we adopted the approach presented by Šmilauer and Lepš [7] using the multivariate ordination methods in Canoco 5.12, starting from the inflated data table. We partitioned gamma diversity (small mammal diversity of forest types) into the alpha (i.e., local diversity at the level of forest plots) and beta (i.e., diversity among forest plots) components. However, beta diversity was given in our dataset, not only by the species turnover among the forest plots of a forest type, but also by the year-to-year changes in community composition, and we assumed that the amplitude of these changes might vary among forest types. Therefore, we considered these two sources of variation separately, as spatial and temporal beta diversity. Total beta resulted from the explained variation in the constrained analysis with plot ID as predictor and beta time from the analysis with year as predictor. Beta space was calculated as total beta minus beta time. Because the design was not orthogonal, there was an overlap between the two beta components (spatial and temporal) included in the beta space. Similarly, at the regional scale, we partitioned the epsilon diversity (small mammal diversity of Romanian forests) into the alpha, spatial beta, temporal beta, and delta (i.e., diversity among forest types) diversities. The delta diversity was calculated from the explained variation of the analyses with forest type as predictor.

## 3. Results

### 3.1. Trapping Results and Elevational Differences

During the 19 years of study, we trapped 1913 individuals of 20 species, the rodents being dominant (Table A1). Five species were singletons or doubletons (species represented by one or two captured specimens), and two of these (*Neomys anomalus* and *Arvicola terrestris*) were not included in the data set of the randomly selected transects, which comprises 860 individuals of 17 species (Table 1). *Microtus levis* was represented by 10 individuals, but they were captured in only one forest, during three of the nine surveys, so this species was also not included in the selection. The most numerous species in the reduced data set was *A. flavicollis*, with 489 individuals (56.9% of the captured individuals, with the standard error of the percentage—SE = 1.7%), followed by *M. glareolus*, with 162 individuals (18.8%, SE = 1.3%) and *A. agrarius* with 92 (10.7%, SE = 1%). In lowland forests, abundance was significantly higher for *A. agrarius* (W = 2035.5, $p < 0.001$), with a mean abundance of 5.45 ind./100 TN (95% confidence interval—CI = 2.01, 8.9) in lowlands and 1.23 ind./100 TN (95% CI = $-0.4$, 2.86) in mountains, and *A. flavicollis* (W = 1775.5, $p = 0.038$), with 13.66 ind./100 TN (95% CI = 7.89, 19.42) in lowlands and 8.45 ind./100 TN (95% CI = 3.99, 12.91) in mountains. *Microtus arvalis* was captured only in some lowland forests, with 1.68 ind./100 TN (95% CI = $-0.07$, 3.44). *Myodes glareolus* and *S. araneus* were more abundant in mountains than in lowlands. The mean abundance of *M. glareolus* in lowlands was 1.14 ind./100 TN (95% CI = 0.01, 2.26) and in mountains 5.4 ind./100 TN (95% CI = 2.6, 8.21), the difference being significant (W = 1060, $p = 0.005$). The mean abundance of *S. araneus* in lowlands was 0.23 ind./100 TN (95% CI = $-0.01$, 0.48) and in mountains 1.04 ind./100 TN (95% CI = 0.5, 1.59), the difference being significant (W = 1197.5, $p = 0.023$). Total abundance was significantly higher in lowland forests (W = 1884.5, $p = 0.008$), with 23.6 ind./100 TN (95% CI = 16.48, 30.8) in lowlands and 17.02 ind./100 TN (95% CI = 10.18, 23.87) in mountains, but not species richness (W = 1629, $p = 0.168$) and the proportion of empty transects ($\chi^2 = 1.81$, df = 1, $p = 0.178$). The effect of elevation on heterogeneity was marginally significant (W = 1062, $p = 0.077$), with the Simpson index being higher in mountains (0.486, 95% CI = 0.401, 0.571) than in lowlands (0.361, 95% CI = 0.265, 0.457).

*3.2. Use of Forest Types by Small Mammal Species*

　　*Apodemus flavicollis* was the most generalist among the forest species (FT Smith = 0.932), using all the studied forest types (Figure 2), followed closely by *M. glareolus* (FT Smith = 0.921), best represented in mixed and spruce forests. Rare species were captured in only one forest type. Among these, *M. musculus* (montane riparian forest), *C. suaveolens*, and *A. uralensis* (poplar plantation) are only accidentally found in forests, being characteristic for open habitats (or anthropic habitats, in case of *M. musculus*).

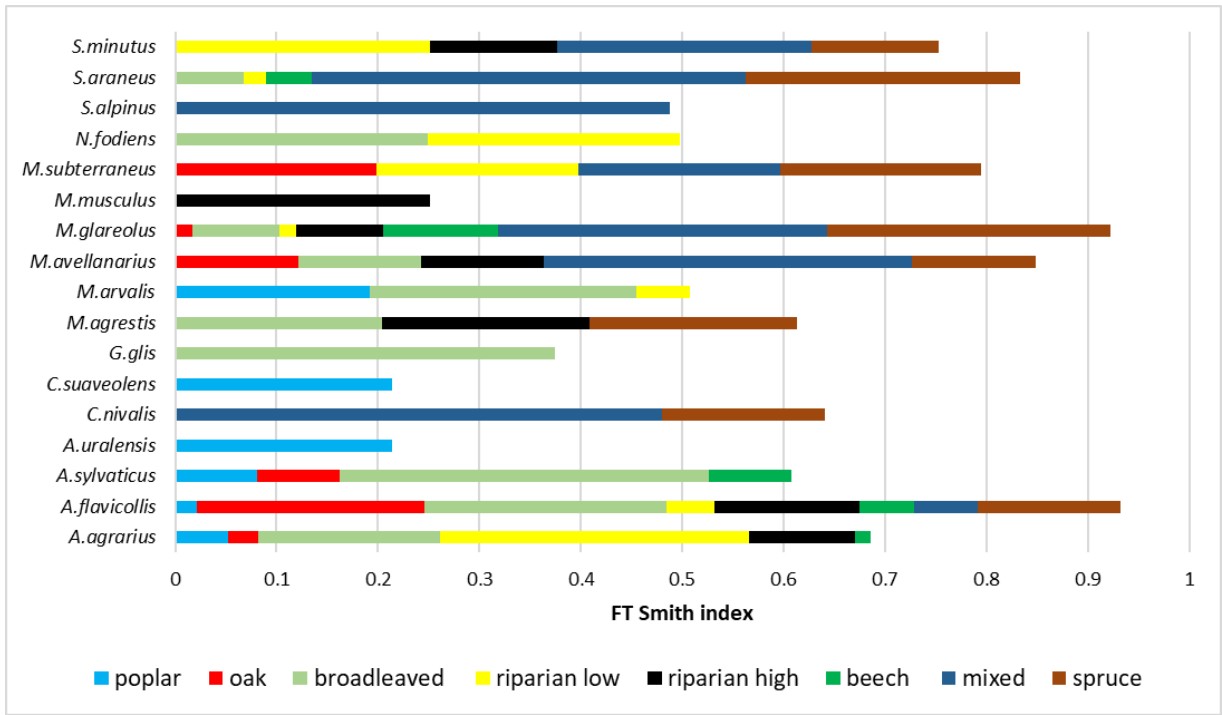

**Figure 2.** Niche width (expressed by FT Smith niche width index) of small mammal species. For each species, relative occurrence in the eight surveyed forest types is given by colour codes. Riparian forests were divided into two distinct types, those of lowlands (riparian low) and those of mountains (riparian high).

　　Forest type had a significant effect (test on all axes, pseudo-F = 2.5, *p* = 0.001) on the structure of small mammal communities, explaining 17.5% (10.5% adjusted) of the variation in species composition. In the ordination space defined by the first (pseudo-F = 1.1, *p* = 0.001) and second (pseudo-F = 0.6, *p* = 0.009) ordination axes, poplar plantations were most distinct from the rest of the forests, with *A. uralensis; C. suaveolens; M. arvalis;* and, to a lesser extent, *A. sylvaticus*, as characteristic species (Figure 3). Poplar plantations explained most of the variation in small mammal community structure (6.2% of the species composition, pseudo-F = 5.9, $p_{\text{adj}}$ = 0.004) (Table 2). *Apodemus uralensis, C. suaveolens,* and *M. arvalis* had significant positive responses to the poplar plantations, while the response of *A. flavicollis* was significantly negative (Figure A1a). Broadleaf and conifer mixed forests had the most distinctive small mammal communities in mountains, with *S. alpinus, C. nivalis, S. araneus,* and *M. glareolus* being the characteristic species (Figure 2). Among these, only *M. glareolus* showed a marginally significant response to this type of forest (Figure A1b), which explained 3.6% of the variation in species composition (pseudo-F = 3.4, $p_{\text{adj}}$ = 0.004) (Table 2). Riparian forests in lowlands had a similarly distinct small mammal community structure (explaining 3.5% of the variation in species composition, pseudo-F = 3.3, $p_{\text{adj}}$ = 0.009), with *A. agrarius* and *N. fodiens* as characteristic species (Figure 2). *Apodemus agrarius* was the only species with significant positive response, while *M. glareolus* showed a significant negative response (Figure A1c).

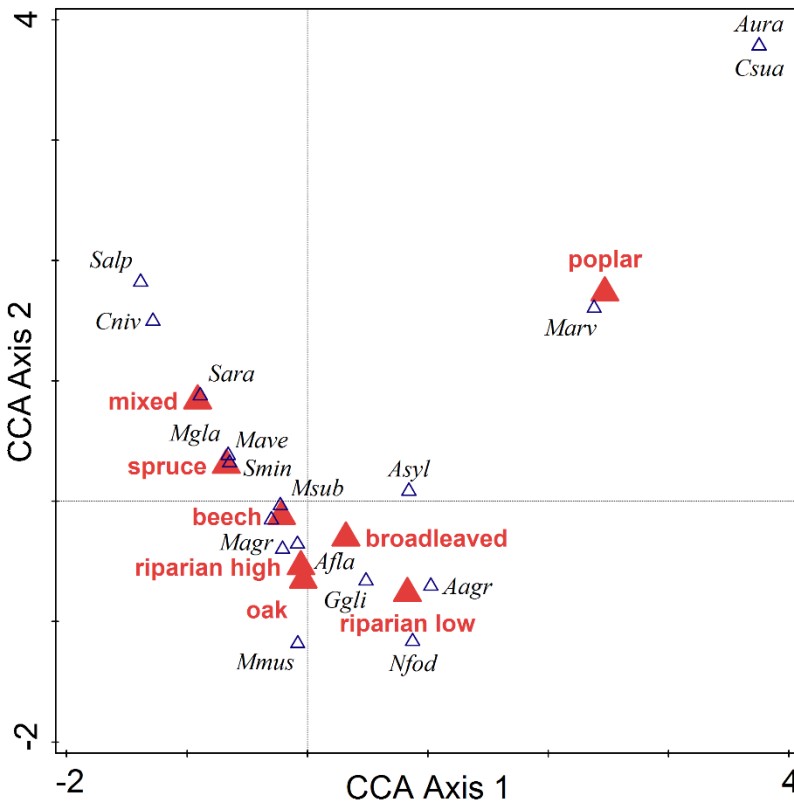

**Figure 3.** Species-habitat biplot diagram from CCA summarising the effect of forest type (red triangles) on species composition. Empty triangles indicate the centroids for the small mammal species. The codes for species are given by the initial of genus and first three letters of species name. Riparian forests were divided into two distinct types, those of lowlands (riparian low) and those of mountains (riparian high).

**Table 2.** Simple effects of the forest type on species composition of small mammals in the study area. Values of the explained variation (explains %), pseudo-F, unadjusted significance ($p$), and significance adjusted using the false discovery rate procedure ($p_{adj}$) are presented.

| Habitat | Explains % | Pseudo-F | $p$ | $p_{adj}$ |
|---|---|---|---|---|
| Poplar | 6.2 | 5.9 | 0.001 | 0.004 |
| Mixed | 3.6 | 3.4 | 0.001 | 0.004 |
| Riparian low | 3.5 | 3.3 | 0.004 | 0.009 |
| Spruce | 1.8 | 1.7 | 0.096 | 0.168 |
| Broadleaved | 1.6 | 1.5 | 0.164 | 0.221 |
| Oak | 1.6 | 1.4 | 0.189 | 0.221 |
| Riparian high | 1 | 0.9 | 0.441 | 0.441 |
| Beech | 0.4 | 0.3 | 0.977 | 0.977 |

In the variation partitioning between habitat and time, the year of survey explained more than twice as much (21.9%, pseudo-F = 1.5, $p$ = 0.016) as habitat type (10%, pseudo-F = 1.6, $p$ = 0.008) of the total variation in community composition, but their mean square was very similar (0.075 for year and 0.076 for habitat), meaning that individual years and habitat types had similar explanatory power (the number of years of survey being more than twice the number of habitat types—17 versus 8).

In the mixed models, species richness was significantly predicted by the diversity of the tree canopy (LR test, $\chi^2$ = 4.24, $p$ = 0.039), while small mammal heterogeneity was predicted by habitat heterogeneity (LR test, $\chi^2$ = 5.02, $p$ = 0.025). Elevation had no additional effect on either of the two measures of diversity.

### 3.3. Species Richness

The rarefaction curves show that species richness was undersampled in most of the forest types, both in lowlands (Figure 4a) and in the mountains (Figure 4b), and this was independent of the number of captured individuals. The greatest difference between the observed (captured) and the estimated (bootstrap estimator) number of species was recorded in the two forest types with the highest number of captured individuals (broadleaved mixed forests—N = 198 individuals, species = 10, bootstrap = 13.73, spruce forests—N = 140 individuals, species = 8, bootstrap = 12.79) (Table 1). At the given spatial sampling effort, species richness was best sampled in some mountain forests, namely, the broadleaved and coniferous mixed forests (N = 120 individuals, species = 8, bootstrap = 8.95) and the beech forests, where the two parameters were equal (5 species), on the basis of the 54 captured individuals. However, significantly increasing the sampling time span yielded higher diversities, even in these forests. Thus, in the mixed forests, where the total number of captured individuals in the original data set was the highest because of the large number of repeated sampling, the total number reached 11, two of them being singletons (*M. subterraneus* and *N. fodiens*) (Table A1). These results suggest the importance of the temporal dimension in the estimation of small mammal species richness.

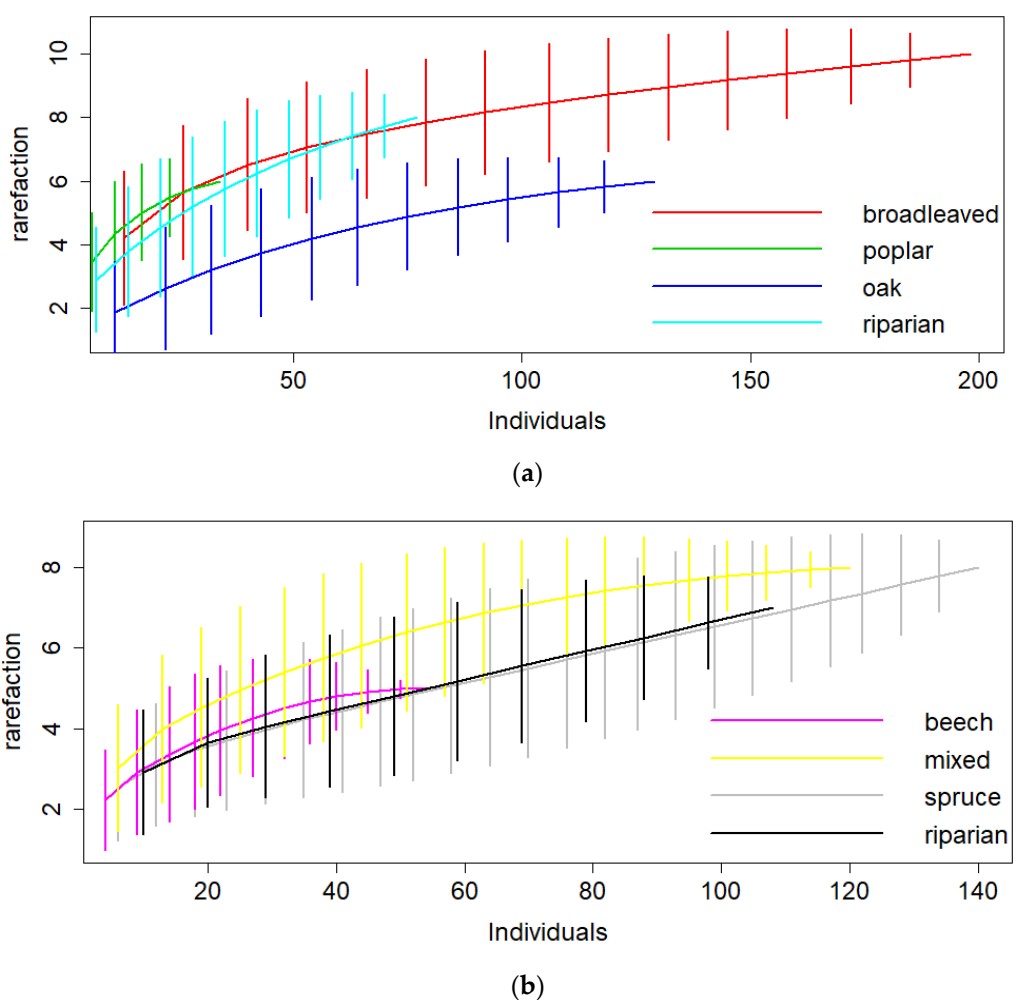

**Figure 4.** Rarefaction curves of small mammal species in the four forest types in (**a**) lowlands and (**b**) mountains. Vertical bars represent the 95% confidence interval of the estimated number of species.

### 3.4. Partitioning of Gamma Diversities

In most forest types, the main source of gamma diversity was represented by the plot-level diversity (alpha) for both species richness and heterogeneity. Beta species richness was

slightly greater in mountain forests, with similar among-plot (spatial beta) and among-year (temporal beta) variation. Beech forests had the lowest alpha diversity, being the only forest type where species turnover (spatial and temporal) had a greater contribution to gamma diversity than local species richness (Figure 5). Heterogeneity (Simpson diversity index) was very variable in lowland forest types, with the lowest value in oak forests, which were clearly dominated by *A. flavicollis*, and the highest in poplar plantations, where densities were low for all species and among-plot variation was highest. In the other forest types, heterogeneity had similar values, with beta diversity resulting mainly from the year-to-year variation in the abundance of dominant species. For broadleaved and riparian lowland forests, the variation in the relative abundance of dominant species was especially low, resulting in very small spatial beta values (Figure A2a).

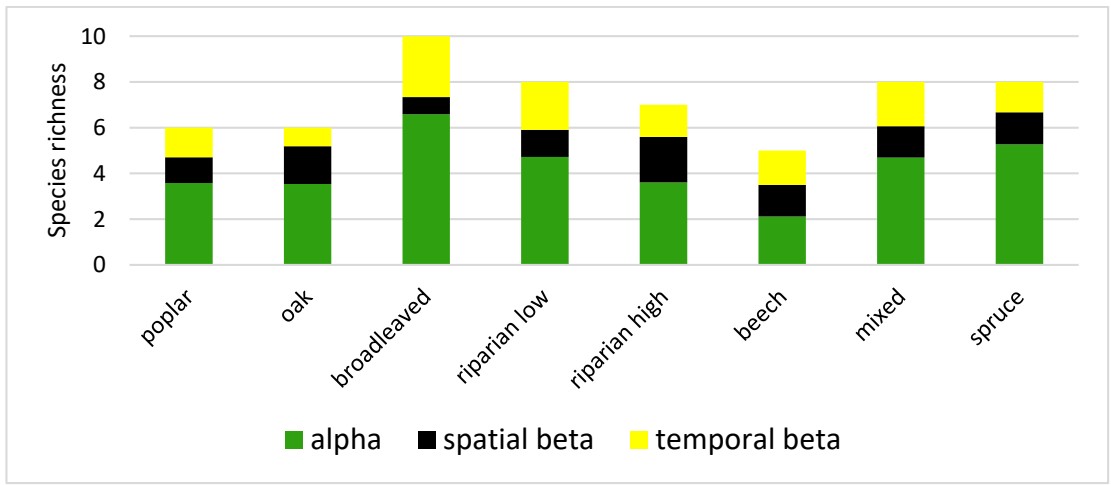

**Figure 5.** Partitioning of species richness of the surveyed forest types (gamma diversities) into local diversity at the level of forest plots (alpha), species turnover among plots of the same forest type (spatial beta), and species turnover among survey years of the same forest type (temporal beta).

### 3.5. Partitioning of Small Mammal Diversity of Forests in Romania

Species turnover among forest types (delta) was low, representing only 5.1% of the total species richness, while the turnover among forest plots of the same habitat type (spatial beta) was much higher (20%) (Figure 6). In case of heterogeneity, variations among forest types (15.8%) were similar to those among forest plots (19.4%), and year-to-year changes were also important (11.5%) (Figure A2b).

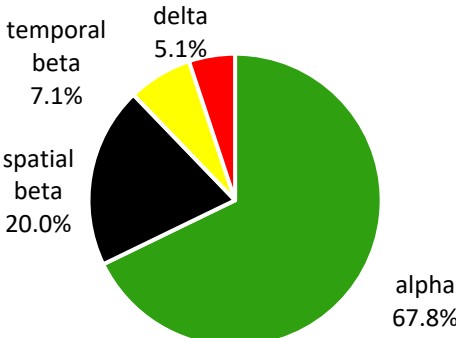

**Figure 6.** Partitioning of species richness of Romanian forests (epsilon diversity) into local diversity at the level of forest plots (alpha), species turnover among plots of the same forest type (spatial beta), species turnover among survey years of the same forest type (temporal beta), and the diversity at the level of forest types (delta).

## 4. Discussion

We surveyed small mammal communities in eight forest types across Romania, a country with low intensity of forest management and a high percentage of natural forests compared to other central European countries. This is the first study giving an overview of habitat type use and diversity of small mammals in forests of this part of Europe.

Small mammal abundance in the surveyed forests was significantly higher in lowlands than in the mountains. This is caused by the harsher environmental conditions at high elevations, especially by the cooler climate, as well as the lower quantity of available energy resulting in fewer resources available for consumers, therefore causing a lower carrying capacity of mountain forests. Species richness, on the other hand, was similar at low and high elevations. This might be related to the mid-domain effect, described for small mammals by McCain [24] in the Neotropics, where species richness is highest in sites at intermediate elevations, where natural conditions are considered to be optimal [25], and where elevational ranges of several lowland and mountain species overlap. The mid-domain effect in small mammals was also documented in Central Europe, with the highest species richness values observed at 500 to 700 m elevations [26,27]. Among the dominant species, *A. agrarius* was characteristic for lowland forests, prevailing in riparian habitats, while *M. glareolus* was more abundant in mountain forests, although it was a common presence in most lowland forests as well (except for poplar plantations), similarly to other lowland forests of Central Europe [28].

The three dominant species (*A. flavicollis*, *M. glareolus*, *A. agrarius*) represented 86.4% of all the captured individuals, which is in accordance with Schröpfer [29], who stated that over 75% of individuals of small mammal communities in Europe are represented by only three species. *Apodemus flavicollis* was the most abundant species, representing more than half of the captured individuals, but also the most widespread, exploiting all the habitat resources (types), but having marginally significantly lower abundances in the mountains, probably because of its stronger population fluctuations at higher elevations [30]. At large scale, in fragmented landscapes, *A. flavicollis* is a forest specialist [31,32], but in forested landscapes, it acts as a habitat generalist, being able not only to exploit all wooded habitats but also to dominate any small mammal forest community, alone or in combination with other species. The generalist character of *A. flavicollis* in forests was also revealed by studies conducted in other parts of its geographical range. In the lowland forests of the South Moravian rural landscape, *A. flavicollis* was dominant, found in all forests, usually with the highest dominance (except for some monocultures), and showed the weakest response to forest characteristics [28]. Although *M. glareolus* was not found in one of the forest types (poplar plantations), because of its much more stable population dynamics [30], it had only a slightly narrower niche compared to *A. flavicollis*.

Forest habitats can usually be divided into early successional forests with dense herb layer and no fruiting trees (plantations) and high forest stands with closed tree canopy, fruiting trees, and sparse herb layer [28]. In our study, community composition was most distinct in poplar plantations, which, although not young, had characteristics closest to early successional forests (i.e., open canopy and dense herb layer). Our studied poplar plantations were defined by the presence of open field specialists such as *M. arvalis* [31], *C. leucodon*, and *A. uralensis*. *Crocidura* shrews were also reported from young oak plantations (a habitat type missing from our dataset) but with lower densities than in poplar plantations [28]. In terms of overall density, species richness, and species composition, plantations are poorer quality habitats than natural or semi-natural forests and they may not function as forests [33]. Small mammal diversity and abundance are often significantly higher in surrounding forests and scrublands than plantations [34]. In plantations, species composition is a mixture of open land (crop and grassland) and forest species that is unique compared to other nearby habitats and does not resemble that of either grasslands or forests, with within-plantation habitat quality and plantation vegetation heterogeneity being important determinants of occupancy [33]. One way to increase habitat heterogeneity is by maintaining a diversity of plantation ages within the complex, resulting in enhanced

small mammal species diversity [35]. Compared to plantations elsewhere, we found similarly low abundances of small mammals in the poplar plantations, but higher species richness and heterogeneity. This is because the species pool in these plantations depends not only on habitat characteristics but also on the forest history and connectivity with other forests. In Romania, poplar plantations are often established along rivers, connected with the riparian forests, hence the presence of highly mobile forest specialists (*A. flavicollis*). However, the abundance and composition of small mammal communities depend on the type of plantations, implemented management, and production cycle. The latter is mostly linked to the plantation's temporal heterogeneity in structure [36,37]. Therefore, our results reflect only the situation in mature poplar plantations and cannot be generalised to other types of plantations.

Regarding the other types of forest, they represented a continuum, with lowland riparian forests as one of the extremes and mixed forests as the other. The observed species richness was associated with the diversity of tree canopy, therefore being highest in the mixed forests. Spruce forests, on the other hand, had the second-highest estimated species richness. This is in contrast with other parts of Central Europe, where spruce forests are poor habitat for small mammals [38]. This is because, unlike in most of Central Europe, where coniferous forests are found well below their natural elevational range and are intensively managed, being usually represented by spruce monocultures [39], in Romania, spruce forests are located mostly within their natural range, with only a few plantations within the range of the beech or below. As a result, spruce forests in Romania are diverse in undergrowth vegetation, with different shrub and herb species adapted to various microhabitat conditions and rocky outcrops that increase habitat heterogeneity, associated with high small mammal diversity [40]. The sink effect of spruce lowland monocultures on small mammals was shown by Suchomel et al. [38], who found that mixed old forest stands surrounded by spruce monocultures of various ages were not able to sustain high diversity of small mammals despite their favourable habitat conditions.

In our study, habitat heterogeneity was significantly related not to species richness but to heterogeneity, with the homogenous environments allowing one species (the best adapted to those specific conditions) to develop dense populations and dominate the community.

In this study, we evaluated small mammal diversity in the forest types best represented in Romania and Central Europe in general. Further research should also focus on other forest types that are still rather uncommon in Romania but may expand in the future, such as short-rotation woody crops or plantations of allochthonous trees (black locust—*Robinia pseudacacia*, honey locust—*Gleditsia triachantos*) that are of economic interest, or forests where invasive species tend to replace the native ones. In the western United States, the riparian habitats dominated by the invasive salt cedar (*Tamarix* spp.) were found to have lower small mammal diversities compared to habitats with a mix of *Tamarix* and native trees, with rare and specialist species being more impacted by non-native vegetation [41].

Most papers examining beta diversity focus on spatial patterns of diversity and their drivers at various scales (e.g., [42–45]) or on the reduction of beta diversity as an effect of species loss and homogenisation of both habitats and faunas as the result of changes in the land use and biological invasions (e.g., [46–48]). From our study, we have learned that beta diversity of small mammals also has an important short-term temporal component. In some forest types, year-to-year species turnover was higher than the spatial, among-plots turnover. While spatial beta diversity is caused by the differences in habitat characteristics among forests of the same type, and at a lesser extent by interspecific relationships, especially competition [32], the underlying mechanisms of temporal beta diversity are mainly related to change in environmental conditions, as the result of weather and human interventions, but also to intrinsic factors involved in population dynamics.

Our results from the forest plot surveyed 60 times showed that long-term monitoring can yield observed species richness that exceeds richness estimated based on single samplings, even in multiple sites, revealing the real diversity potential of specific habitat

types. Therefore, for the assessment of small mammal diversity in temperate forests and the conservation value of different forest types, communities need to be surveyed each year for an extended time (at least 4–5 years).

We showed the importance of year in estimating the richness and heterogeneity of small mammal communities. However, because various species show different annual variations in abundance, sampling other seasons (winter, spring) would allow a more detailed image of the temporal diversity patterns by decomposing beta time in beta season and beta year.

In this paper, we aimed to partition beta diversity into its spatial and temporal components, but there are also other ways to decompose beta diversity. Legendre and De Cáceres [49] proposed partitioning the total variation in the communities, synthesised in a species-by-site matrix (total beta), into the contributions of individual sites and species. The local contributions to beta diversity indicate the biological uniqueness of each site and may be used to identify sites with an unusual combination of species with high conservation value or degraded sites in need of restoration [3]. Recently, this approach has proved to be useful in examining and comparing the ecological uniqueness among different sites, revealing the regional scale current status of mammal diversity in the Neotropics [50].

The partitioning of beta diversity into species replacement (turnover) and richness difference (nestedness) proposed by Baselga [51] enables the disentangling of the underlying mechanisms that generate large-scale diversity. Species turnover is driven mainly by environment gradients and competition, while nestedness reflects local abiotic conditions that may result in local absences of species [3]. Partitioning beta diversity into turnover and nestedness may also reveal patterns of homogenisation that cannot be revealed using the classical approach of similarity analysis [46].

Taxonomic diversity is, however, only one of the numerous facets of biological diversity. The evaluation of other aspects (functional, phylogenetic, niche-based) of diversity at various scales will further contribute to the knowledge of patterns that govern small mammal distribution in temperate forests and their underlying mechanisms.

## 5. Conclusions

Small mammal communities in temperate forests of Romania are dominated by *A. flavicollis*, a forest generalist having the widest habitat niche, which was captured in all types of forests. The other prevailing rodents were *M. glareolus*, characteristic of mountain forests, and *A. agrarius*, most abundant in lowland riparian forests. Among the shrew species, *S. araneus* was the most abundant, found mainly in mountain forests. As hypothesised, poplar plantations had the most distinctive composition of small mammal communities, comprising not only forest species but also open habitat specialists, such as *M. arvalis* or *C. suaveolens*, but having low population densities, which results in the highest heterogeneity. Overall, heterogeneity had a temporal component that was more important than the spatial one because of the strong fluctuation in population density of dominant rodents (especially *A. flavicollis*). Species richness also had an important temporal component. Our results suggest that ignoring the time dimension in the survey of small mammal communities may lead to underestimating species richness, both at the local and regional scales.

**Author Contributions:** Conceptualisation, A.L., A.M.B. and I.S.; methodology, A.L., A.M.B. and I.S.; software, A.M.B.; validation, A.L., A.M.B. and I.S.; formal analysis, A.M.B.; field investigation, A.L., A.M.B. and I.S.; resources, A.M.B.; data curation, A.L. and A.M.B.; writing—original draft preparation, A.M.B.; writing—review and editing, A.L., A.M.B. and I.S.; visualisation, A.M.B.; supervision, A.M.B.; project administration, A.M.B.; funding acquisition, A.M.B. All authors have read and agreed to the published version of the manuscript.

**Funding:** A.M.B. acknowledges the project financed by Lucian Blaga University of Sibiu and Hasso Plattner Foundation research grants LBUS-IRG-2020-06.

**Institutional Review Board Statement:** All aspects of trapping and animal handling complied with EU Council Directive 86/609/EEC on experimental use of animals. Trapping within the protected area was done at the invitation of the Administration of Retezat National Park following the protocols on trapping and animal handling developed and approved by the Scientific Council of Retezat National Park. Subsequently, the working protocol was also approved by the Biomedical Research Ethics Commission of Lucian Blaga University of Sibiu (approval 9/26.05.2021).

**Informed Consent Statement:** Not applicable.

**Data Availability Statement:** The data sets used for the presented analyses are included in the paper.

**Acknowledgments:** The authors thank Attila D. Sándor, Erika Stanciu, Călin Hodor, Zoran Acimov, and the Administration of the Retezat National Park for the invitation to take part in the faunistical inventory program conducted in the park; Mihai Vasile, Marius Drugă, Iounț Bordea, and Alex Nicoară for their assistance in the field; Aurelian Bordei for the small mammal data from two poplar plantations; and the two anonymous reviewers for their constructive comments and suggestions that helped improve the article.

**Conflicts of Interest:** The authors declare no conflict of interest. The funders had no role in the design of the study; in the collection, analyses, or interpretation of data; in the writing of the manuscript; or in the decision to publish the results.

## Appendix A. Description of Surveyed Forest Types

### *Appendix A.1. Lowland Forests*

Natural forests in Romanian lowlands are dominated by oak species: *Quercus robur*, *Q. pedunculiflora*, *Q. pubescens*, *Q. cerris*, and *Q. frainetto* in the plains and *Q. petraea*, *Q. dalechampii*, and *Q. polycarpa* on hills. Oak forests usually do not have closed canopies, allowing the development of dense understory vegetation, comprising various shrub species (*Crataegus monogyna*, *Prunus spinosa*, *Ligustrum vulgare*, *Evonymus europaeus*, *Cornus sanguineus*, *C. mas*). Natural regeneration of oaks is difficult, and often the shrub layer is formed mainly of hornbeam (*Carpinus betulus*) saplings and young trees, which, in most managed forests, are consistently cut and usually left in place, resulting a great amount of coarse woody debris.

The mixed broadleaved forests may have various compositions, depending on site characteristics. Hornbeam, elm (*Ulmus procera*, *U. minor*), linden (*Tilia cordata*, *T. platyphillos*), ash (*Fraxinus excelsior*), and maple (*Acer campestre*, *A. platanoides*) may be dominant or present in various proportions in the canopy of these forests. The shrub species are shared with the oak forests and are usually well represented. Hazel (*Corylus avellana*) may also be present.

At low elevations, on flat grounds, especially in the vicinity of watercourses, poplar plantations were established using either native species (*Populus alba*, *P. nigra*) or commercial hybrids (*P. canadensis*). These plantations lack the tall shrub layer almost completely, but dewberry (*Rubus caesius*) may be abundant. The herb layer is very well developed, especially in young plantations, and it is mainly composed of grass species. Because the trees are young, the coarse woody debris is sparse, represented mostly by branches torn by storms.

Along rivers, riparian forests cover a narrow stripe of usually 10–15 m width, or even less. The dominant trees are willows (*Salix alba*, *S. fragilis*, *S. triandra*), and in the hills, the black alder (*Alnus glutinosa*). The tree canopy is usually reduced, and the shrub layer may be well developed and diverse, including mostly *Cornus sanguinea* and *Evonymus europaeus*, and the herb layer is tall and diverse, usually dominated by nitrophilous forbs. *Rubus caesius* is often abundant, forming thickets that represent an important microhabitat in these forests. The various layers in these forests are connected by lianas (*Humulus lupulus*, *Clematis vitalba*) that sometimes form thick covers on trees and shrubs. Riparian forests shelter the most numerous allochthonous plant species (trees—*Acer negundo*, shrubs—*Amorpha fruticosa*, lianas—*Echinocystis lobata*, herbs—*Impatiens balsaminifera*) among all forests in Romania. Lowland riparian forests are subjected to high anthropic pressure, most

evident near localities, where local people cut wood for heating and deposit waste on the riverbanks. Part of the forests are completely clearcut and replaced by crops, altering the continuity of the riparian habitats.

*Appendix A.2. Montane Forests*

Riparian forests in mountains differ from those in the lowland mostly by their connectivity with other forests, situated on the slopes. Montane riparian forests are dominated by grey alder (*Alnus incana*) and goat willow (*Salix caprea*). The tree canopy cover is sparse, the herb layer is rich and high, and the substrate is rocky.

At low elevations there are mainly beech forests dominated by *Fagus sylvatica*. Sometimes other broadleaved trees are also present in the canopy, such as the sycamore maple (*Acer pseudoplatanus*) or the wych elm (*Ulmus glabra*). Beech forests usually have a very dense tree canopy; therefore, the shrub and herb layers are sparse.

Mixed forests, composed of differing proportions of beech and Norway spruce (*Picea abies*) with scattered silver fir (*Abies alba*) and sycamore maple, are characteristic of higher elevations. Mixed forests have a wide range of habitat characteristics, depending mostly on the tree canopy structure and the forest management.

A spruce forest belt reaches up to the timberline, which usually is present at elevations between 1600 and 1800 m, depending on slope exposition and other geomorphlogical characteristics of the site. The shrub and herbaceous layers vary greatly in spruce forests, the understory being composed mainly of spruce saplings, often with blueberry (*Vaccinium myrtillus*) bushes and a thick moss layer. The tree canopy cover of spruce forests decreases near the timberline, but the herbaceous layer is also reduced due to the rocky outcrops and surfacing stones. At timberline, mountain-ash (*Sorbus aucuparia*), stone pine (*Pinus cembra*), and juniper (*Juniperus communis*) shrubs are interspersed among dwarf spruce trees.

This natural elevational succession of forest habitats is sometimes altered by temperature inversions or past logging and reforestation, which artificially lowered the lower limit of spruce forests.

**Table A1.** Small mammal trapping results in all the transects in the eight surveyed forest types, some with multiple temporal replicates in various years.

| Habitat Type | Poplar Plantation | Oak Forest | Mixed Broadleaved Forest | Lowland Riparian Forest | Montane Riparian Forest | Beech Forest | Mixed Forest | Spruce Forest | Total |
|---|---|---|---|---|---|---|---|---|---|
| No of transects | 6 | 19 | 28 | 35 | 17 | 26 | 83 | 50 | 264 |
| *Apodemus agrarius* | 7 | 14 | 30 | 169 | 14 | 2 | 5 | 0 | 241 |
| *Apodemus flavicollis* | 11 | 149 | 167 | 49 | 89 | 93 | 253 | 128 | 939 |
| *Apodemus sylvaticus* | 2 | 2 | 10 | 7 | 0 | 2 | 0 | 0 | 23 |
| *Apodemus uralensis* | 1 | 0 | 0 | 0 | 0 | 0 | 0 | 0 | 1 |
| *Arvicola terrestris* | 0 | 0 | 0 | 2 | 0 | 0 | 0 | 0 | 2 |
| *Chionomys nivalis* | 0 | 0 | 0 | 0 | 0 | 0 | 26 | 1 | 27 |
| *Glis glis* | 0 | 0 | 5 | 0 | 0 | 0 | 5 | 0 | 10 |
| *Microtus agrestis* | 0 | 0 | 1 | 0 | 1 | 0 | 0 | 5 | 7 |
| *Microtus arvalis* | 11 | 0 | 15 | 5 | 0 | 0 | 0 | 0 | 31 |
| *Microtus levis* | 0 | 0 | 0 | 10 | 0 | 0 | 0 | 0 | 10 |
| *Microtus subterraneus* | 0 | 1 | 0 | 2 | 0 | 0 | 1 | 1 | 5 |
| *Mus musculus* | 0 | 0 | 0 | 0 | 1 | 0 | 0 | 0 | 1 |
| *Muscardinus avellanarius* | 0 | 2 | 1 | 0 | 1 | 0 | 8 | 1 | 13 |
| *Myodes glareolus* | 0 | 10 | 21 | 8 | 18 | 56 | 243 | 105 | 461 |
| *Crocidura suaveolens* | 2 | 0 | 0 | 0 | 0 | 0 | 0 | 0 | 2 |
| *Neomys anomalus* | 0 | 0 | 0 | 0 | 0 | 1 | 0 | 0 | 1 |
| *Neomys fodiens* | 0 | 0 | 2 | 1 | 0 | 0 | 1 | 1 | 5 |
| *Sorex alpinus* | 0 | 0 | 0 | 0 | 0 | 0 | 3 | 4 | 7 |
| *Sorex araneus* | 0 | 0 | 3 | 2 | 3 | 7 | 41 | 54 | 110 |
| *Sorex minutus* | 0 | 0 | 0 | 2 | 1 | 1 | 8 | 5 | 17 |
| Total individuals | 34 | 178 | 255 | 257 | 128 | 162 | 594 | 305 | 1913 |

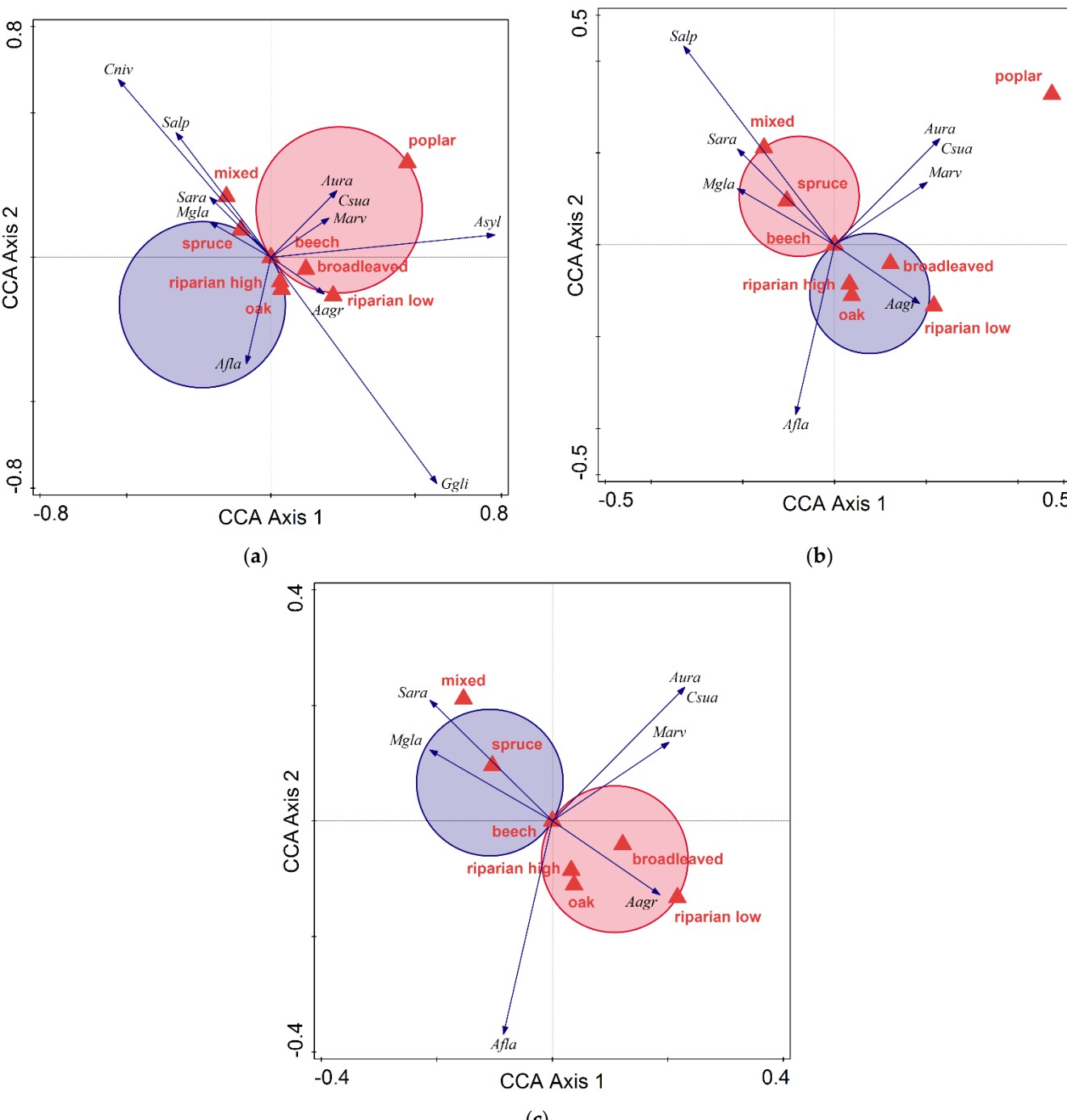

**Figure A1.** The t-value biplots, displaying the first two CCA axes with Van Dobben circles drawn for the forest types with a significant effect on species composition: (**a**) poplar forest, (**b**) mixed forest, (**c**) riparian forest in lowland. Species indicated by arrows ending inside the pink circle show a positive significant response to that forest type at $p = 0.05$, having increased relative abundance, while the species indicated by arrows ending inside the blue circle show a negative response, significant at $p = 0.05$, the response being stronger when arrows are shorter. For the other species, relative abundance was not significantly related to forest type.

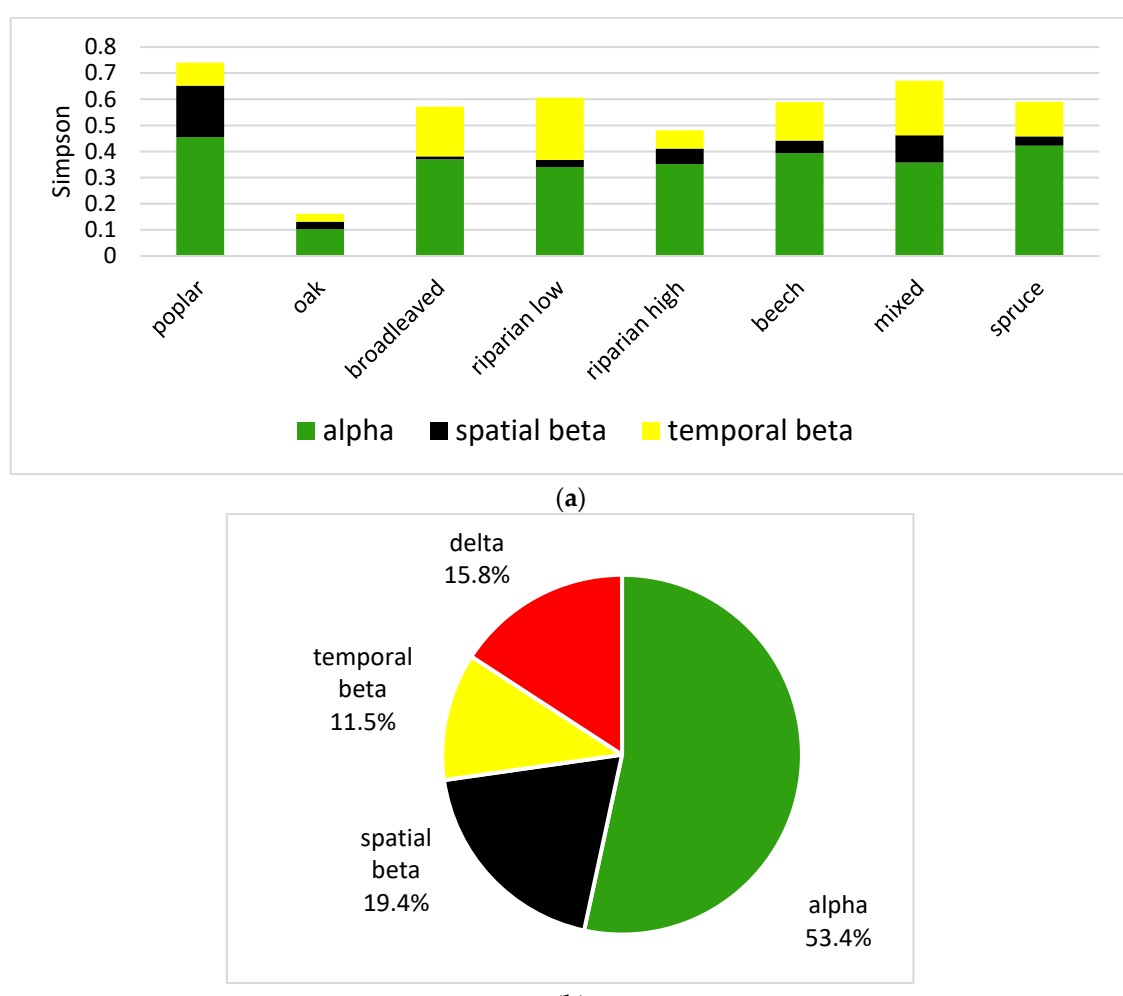

**Figure A2.** Partitioning of heterogeneity (expressed as Simpson index) (**a**) of the surveyed forest types (gamma diversities) into local diversity at the level of forest plots (alpha), species turnover among plots of the same forest type (spatial beta) and species turnover among survey years of the same forest type (temporal beta) (**b**) of Romanian forests (epsilon diversity) into local diversity at the level of forest plots (alpha), species turnover among plots of the same forest type (spatial beta), species turnover among survey years of the same forest type (temporal beta), and the diversity at the level of forest types (delta).

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
