# Peer review of "Small Mammals in Forests of Romania: Habitat Type Use and Additive Diversity Partitioning"

_forests, doi:10.3390/f12081107_

Round 1

Reviewer 1 Report

This study tries to identify patterns in small mammal diversity in Romanian forests. A large dataset was used for this study providing interesting information on a highly biodiverse area in Europe where little research has been published.

The manuscript is well written. I do, however, find the results section very heavy and I am not convinced that all the information and figures are useful to answer the aims of the study. I also think that the first section in the introduction is very long and technical and its content is probably not all essential for this study. I think it can overall be a bit more succinct.

Other comments:

11 – Remove the examples for the abstract.

26-27 – I'm not sure if this is the best way to finish the abstract.

34 – A reference is needed

80 – “have a also” typo

82 – “From an economic”

87-88 – confusing sentence

95 – It is not clear what the 1.4% represent as there are percentages beforehand. It is also not clear if there’s a reason for mentioning the southern Carpathians. A bit of clarity would be useful.

99 – “…there has not been any studies performed at a regional level…”

2.1. Description of forests in Romania – This section is interesting but I’m not sure how it fits in the manuscript as it is not essential to understand the methods. Maybe it could be in the supplementary information? This would reduce the text load in the methods.

190 – “In most forest plots,”

190 – What was the logic behind randomly surveying multiple times. Was it actually randomised or were they surveyed multiple times for reasons that are not linked to this study? Some precision is needed I think.

200 – This could be where the supplementary material about different habitat types is cited.

A map of the study area with sites would be useful.

276-278 – This sentence feels more like a discussion

295 – delete “by far”

439-441 Improve sentence as it is not very clear.

513 – Maybe change “possibilities” to “ways” or “methods”

The manuscript needs a better conclusion summarising the results and discussion of the study. It currently doesn't feel like it finishes.

Author Response

We are grateful for the fast and comprehensive review of our manuscript by the two reviewers. We went carefully through all the comments and suggestions and hopefully we were able to address them all in a satisfactory manner to enhance the clarity and soundness of the manuscript. All the comments and suggestions were responded to in the Point-by-point response to reviewers included below and the necessary changes were made in the manuscript.

Point-by-point responses to reviewers

# Reviewer 1

This study tries to identify patterns in small mammal diversity in Romanian forests. A large dataset was used for this study providing interesting information on a highly biodiverse area in Europe where little research has been published.

The manuscript is well written.

Thank you for your appreciation.

I do, however, find the results section very heavy and I am not convinced that all the information and figures are useful to answer the aims of the study.

We have moved to Supplementary Material some of the results – figures 3, 5b, 6b, which are less essential to understanding the conclusions of the study. If further changes are needed to improve the manuscript, we are happy to make them.

I also think that the first section in the introduction is very long and technical and its content is probably not all essential for this study.

Indeed, most part of the first section is technical and does not really belong to the introduction, but we consider it essential for understanding the method we used for diversity decomposition, especially for those who are not familiar with the multivariate approach, therefore we moved that paragraph to the data analysis section.

I think it can overall be a bit more succinct.

We did a series of changes to the content of the paper and hopefully we managed to address this observation. If suggested, we will make further changes to make the manuscript more digestible.

Other comments:

11 – Remove the examples for the abstract.

We removed the examples for the abstract.

26-27 – I'm not sure if this is the best way to finish the abstract.

We changed the last sentences of the abstract to: ”Because of strong fluctuations in population density of dominant rodents, the temporal component of beta heterogeneity was larger than the spatial one, but species richness also presents an important temporal turnover. Our results show the importance of the time dimension in the design of the surveys aiming at estimating diversity of small mammal communties, both at local and regional scale”. Hopefully this is more appropriate.

34 – A reference is needed

We added a reference.

80 – “have a also” typo

We have corrected the typing mistake.

82 – “From an economic”

We have changed “the” with “an”.

87-88 – confusing sentence

We simplified it to “the rest are forests in hilly areas”.

95 – It is not clear what the 1.4% represent as there are percentages beforehand.

Yes, plantations of allochtonous species and old-growth forests have the same percentage. We added “1.4% of all Romanian forests”

It is also not clear if there’s a reason for mentioning the southern Carpathians. A bit of clarity would be useful.

Indeed, there is no obvious reason for mentioning the southern Carpathians, so we added as explanation “Southern Carpathian Mountains, where most of our montane study sites were located”

99 – “…there has not been any studies performed at a regional level…”

We made the suggested change.

2.1. Description of forests in Romania – This section is interesting but I’m not sure how it fits in the manuscript as it is not essential to understand the methods. Maybe it could be in the supplementary information? This would reduce the text load in the methods.

Indeed, this is a good sugestion. We moved the whole description of forest types in the appendix.

190 – “In most forest plots,”

We added the comma.

190 – What was the logic behind randomly surveying multiple times. Was it actually randomised or were they surveyed multiple times for reasons that are not linked to this study? Some precision is needed I think.

The schedule of the multiple surveys is not linked to this study. We added as explanation at the beginning ot the data analysis paragraph: “The dataset used in this study comes from different small mammal inventory programs, therefore the design is not balanced.”

200 – This could be where the supplementary material about different habitat types is cited.

We added a citation of the Appendix with habitat description.

A map of the study area with sites would be useful.

We included a map with the position of the researched forests.

276-278 – This sentence feels more like a discussion.

Indeed, therefore we removed it completely.

295 – delete “by far”

We deleted it.

439-441 Improve sentence as it is not very clear.

We rephrased the sentence to: “Based on overall density, species richness and species composition are poorer quality habitats than natural or semi-natural forest and they may not function as forests”.

513 – Maybe change “possibilities” to “ways” or “methods”

We changed “possibilities” to “ways”.

The manuscript needs a better conclusion summarising the results and discussion of the study. It currently doesn't feel like it finishes.

Indeed, the previous version of the manuscript did not have a proper conclusion paragraph. We have now added one.

Reviewer 2 Report

The manuscript entitled “Small mammals in forests of Romania: habitat type use and additive diversity partitioning” by Lazar et al. aimed at assessing diversity and richness pattern of small mammals in Romania forest patches. Overall, this manuscript tackles a very interesting and important issues, by generating data that may also facilitate managers to implement efficient conservation plans. It is overall a well structure and well written manuscript, which is in my opinion an important approach to understand how forest types shape small mammals communities, and diversity metrics. In general, I agree with the methodological approach, although some details are missing and extra analysis may shade some light into issues that authors discuss, but not properly test (se bellow). Some parts of the discussion should be reviewed. Detailed comments are listed below:

Abstract

  1. Page 1, Line 10 – You should add some examples of the roles of small mammals may have forests.;

Introduction

  1. Page 2, Line 78 – Please explain what you mean by “caching behavior” and how this influence seed dispersion ability;
  2. Page 2, Line 84 – Kamler et al (2012) is listed in the reference section as Kamler et al-. 2011;
  3. Page 2, Line 44 – Change “espaloished” to “established”;

Methods

  1. Page 3, Line 134 – Change to “10-15m width”;
  2. Page 4, Lines 175-177 – All forest transects were implemented in each year? If not, how did you cope with the probable year’s variation in small mammal’s density? Animals were sampled only in the warm season. Why did you select this season? No seasonal variation in density was considered? Could the patterns showed by your data vary if the entire year was sampled?;
  3. Page 4, Lines 182-183 – You only used 2/3 days for trapping. I believe having traps only active for 2 days will limit you strapping success, especially because the first day usually correspond to a lower trapping success. With a variation of 2 or 3 trapping night, the trapping effort varied. How did you cope with this variation that can affect the richness and diversity patterns?;
  4. Page 4, Lines 190-196 – Again, authors state that some forest patches were sampled more than once, while others just one time. Together with the variation in the number of trapping days, this results in a variation of the trapping effort between patches (See next comment on the trapping effort), which is an import issue when your try to explain your patterns difference between patches with the patches structure and altitudinal location. This should be clearly explained;
  5. Pages 4-5, Lines 202-210 – Capture index – When calculating this index did you accounted for the capture of animals that were trapped more than once? If you have a recapture, the trap is no longer available to capture a new individual (while is occupied by the recapture animal). Thus, the number of traps available for capturing a new animals should be adjusted, by subtracting the number of recaptures to the overall number of trap-nights, to correct the trapping effort. Furthermore, to estimate the abundance of species A you need to correct the sampling effort, because traps that captured animals of species B, C, D,….etc., are not available to capture individuals of species A. Missing to do this will generate a bias linked to an overestimation of the trapping effort, that will increase with the number of recaptures and of captures of individuals from distinct species that the one targeted. You mention using a relative occurrence index. Why not use a relative abundance index instead (e.g. Marques, S. F., Rocha, R. G., Mendes, E. S., Fonseca, C., & Ferreira, J. P. (2015). Influence of landscape heterogeneity and meteorological features on small mammal abundance and richness in a coastal wetland system, NW Portugal. European Journal of Wildlife Research, 61(5), 749–761. doi:10.1007/s10344-015-0952-2)? ;
  6. Page 5, Lines 216-217 – What is the publication year of Ter Braak and Smilaeur? Again, if the trapping effort differed between patches, how did you compare your results taking into consideration this bias?;
  7. Page 5, Lines 228-232 – All patches were sampled every year? If not, how did you cope with individual variation within patches between years?
  8. Page 5, Lines 238-244 – I suggest authors to consider the use of Hill numbers, which seem to be a high robust approach for these objectives (Chao, A., Colwell, R. K., Gotelli, N. J., & Thorn, S. (2019). Proportional mixture of two rarefaction/extrapolation curves to forecast biodiversity changes under landscape transformation. Ecology letters, 22(11), 1913-1922. doi:https://doi.org/10.1111/ele.13322).

Discussion

  1. Page 13, Lines 237-238 – What is the ecological explanation for a high density of Cocidura shrews in plantations?;
  2.  Page 13, Lines 242-243 – Its assumed that plantations support always lower abundances of small mammals than surrounding environments. But this will depend on the type of plantations, implemented management and production cycle. The later if mostly linked to plantation’s temporal heterogeneity in structure (see; Teixeira, D., Carrilho, M., Mexia, T., Köbel, M., Santos, M. J., Santos-Reis, M., & Rosalino, L. M. (2017). Management of Eucalyptus plantations influences small mammal density: Evidence from Southern Europe. Forest Ecology and Management, 385, 25-34. doi:10.1016/j.foreco.2016.11.009; Timo, T. P. C., Lyra-Jorge, M. C., Gheler-Costa, C., & Verdade, L. M. (2014). Effect of the plantation age on the use of Eucalyptus stands by medium to large-sized wild mammals in south-eastern Brazil. iForest - Biogeosciences and Forestry, 8, 108-113. doi:10.3832/ifor1237-008);
  3. Page 13, Line 449 – Here authors compare their data with plantations elsewhere. But to what kind of plantations? Sometimes this, low diversity and richness pattern is not the detected pattern. Depends on the landscape context and the plantations’ type, management and structure.
  4. Page 14, Lines 456-460 – Authors infer that the species richness patterns are linked to tree canopy diversity and habitat heterogeneity. But this effect was inferred not tested. I suggest testing this. Using richness as the dependent variable and the stand size, canopy diversity, location of the stand, habitat heterogeneity, as candidate drivers in a GLM or GLMM approach, you can test these inferred hypothesis;
  5. Page 14, Lines 482-484 – A more detailed conservation and management implication can be derived from these data. I suggest authors to detail why are you suggesting this and be more clear on what is their advise to future researcher to consider in their monitoring scheme to acquire a more realistic data set.
  6. Page 14, Line 499 – Vellend 2006 et al in listed in the reference list as Vellend et al. 2007;
  7. Page 14, Lines 507-511– In this paragraph authors suggest that is important to assess the diversity/richness drivers. I think, as I mentioned earlier, that they can include such approach in the new version of the manuscript.

Author Response

We are grateful for the fast and comprehensive review of our manuscript by the two reviewers. We went carefully through all the comments and suggestions and hopefully we were able to address them all in a satisfactory manner to enhance the clarity and soundness of the manuscript. All the comments and suggestions were responded to in the Point-by-point response to reviewers included below and the necessary changes were made in the manuscript.

Point-by-point responses to reviewers

# Reviewer 2

The manuscript entitled “Small mammals in forests of Romania: habitat type use and additive diversity partitioning” by Lazar et al. aimed at assessing diversity and richness pattern of small mammals in Romania forest patches. Overall, this manuscript tackles a very interesting and important issues, by generating data that may also facilitate managers to implement efficient conservation plans. It is overall a well structure and well written manuscript, which is in my opinion an important approach to understand how forest types shape small mammals communities, and diversity metrics.

Thank you for your appreciation.

In general, I agree with the methodological approach, although some details are missing and extra analysis may shade some light into issues that authors discuss, but not properly test (see bellow). Some parts of the discussion should be reviewed.

We addressed all the comments and suggestions and if changes are still needed to improve our manuscript, we will be happy to make them.

Detailed comments are listed below:

 Abstract

  1. Page 1, Line 10 – You should add some examples of the roles of small mammals may have forests.;

We have added the main roles of small mammals in forests.

 Introduction

  1. Page 2, Line 78 – Please explain what you mean by “caching behavior” and how this influence seed dispersion ability;

We added “because of their caching behaviour, they are important seed dispersers, as they tend to hoard seeds in microsites where the emergence of seedlings would be enhanced and their survival increased because of lower densities of conspecific trees”.

  1. Page 2, Line 84 – Kamler et al (2012) is listed in the reference section as Kamler et al-. 2011;

The paper was published in 2011. We corrected the citation in the text.

  1. Page 2, Line 44 – Change “espaloished” to “established”;

We corrected the typo.

 Methods

  1. Page 3, Line 134 – Change to “10-15m width”;

We made the change.

  1. Page 4, Lines 175-177 – All forest transects were implemented in each year?

No, as we state later, some forest were sampled once (and we added “in different years”), while others were sampled multiple times, but not simultaneously.

If not, how did you cope with the probable year’s variation in small mammal’s density?

In the CCA with habitat as predictor, the effect of year was accounted for by including it as covariate (resulting a partial CCA). This means that in the analysis all the variation that may be explained by the year was removed, and than the effect of habitat was tested on the residuals.

In the decomposition of beta diversity, because the design is not orthogonal, there is an overlap between the two beta components (saptial and temporal), which is included in beta space.

For the estimation of richness and heterogeneity at forest type level, we consider that having randomly selected sites surveyed in different randomly selected years, both with low and high density, we have a representative dataset for these habitat types.

Animals were sampled only in the warm season. Why did you select this season?

As previously states, the dataset from this study originates from various survey programs, with different designs, but which were all conducted during the warm season for accessibility reasons. Only two forest stands (one lowland riparian and one mixed forest) were surveyed also in winter and spring, but, to get comparable data, we excluded these data in the first place, before randomly extracting one sample from each of the two data series.

No seasonal variation in density was considered?

We checked the effect of season on the parameters that we used in analyses, but it was not significant (probably because most trapping was done in August-September), so we did not include it in the final analyses that we presented in the manuscript.

Could the patterns showed by your data vary if the entire year was sampled?;

Yes, because various species show different annual variations in abundance, sampling also other seasons may result in different results, and this may be the focus of another study with a specific design – the decomposition of beta time in beta season and beta year.

We added to the discussion section:

“We showed the importance of year in estimating richness and heterogeneity of small mammal communities. However, because various species show different annual variations in abundance, sampling also other seasons (winter, spring) would allow a more detailed image on the temporal diversity patterns, by decomposing beta time in beta season and beta year.”

  1. Page 4, Lines 182-183 – You only used 2/3 days for trapping. I believe having traps only active for 2 days will limit you strapping success, especially because the first day usually correspond to a lower trapping success. With a variation of 2 or 3 trapping night, the trapping effort varied. How did you cope with this variation that can affect the richness and diversity patterns?;

Unfortunately the uneven trapping effort is one of the main shortcomings of the manuscript, as it relies on a heterogenous dataset originating from various survey programs. And another source of variation in the trapping effort is represented by non-functional traps (disturbed or destroyed by various factors, human or non-human), that is sometimes difficult to control in the field. Therefore, further studies on this aspect should rely on datasets originating from balanced designed surveys. However, even with equal trapping efforts, due to the high amplitude variations in population densities among years, especially in some species, richness will be underestimated in some stands. For good estimates of richness time needs to be considered (evaluation should be based on long-time datasets), and this is one of the conclusions of the study.

In the revised version of the manuscript we also accounted for the difference in trapping effort among habitat types when calculating the species’ niche width, replacing Levins index (which considers only pi – the proportion of the total number of individuals of the considered species that exploit resource i – in our case, which populate habitat type I, with FT Smith index, which includes also ai – the proportion of the resource i in the environment – in our case, the proportion of trapping effort used in forest type i.

  1. Page 4, Lines 190-196 – Again, authors state that some forest patches were sampled more than once, while others just one time. Together with the variation in the number of trapping days, this results in a variation of the trapping effort between patches (See next comment on the trapping effort), which is an import issue when your try to explain your patterns difference between patches with the patches structure and altitudinal location. This should be clearly explained;

Although some forest patches were sampled multiple times, in the analyses we included only one sample (randomly selected) per forest stand (regardless how many times it was surveyed), so there is no variation in the number of samplings per plot, only in the trapping effort, a shortcoming that we addressed beforehand.

  1. Pages 4-5, Lines 202-210 – Capture index – When calculating this index did you accounted for the capture of animals that were trapped more than once?

No, in the original version of the manuscript we did not.

If you have a recapture, the trap is no longer available to capture a new individual (while is occupied by the recapture animal). Thus, the number of traps available for capturing a new animals should be adjusted, by subtracting the number of recaptures to the overall number of trap-nights, to correct the trapping effort.

Yes, we agree. In the revised version of the manuscript we subtracted the number of traps with recaptures from the overall number of trap-nights and redid the analyses concerning the differences between lowlands and mountains. In the other analyses we used the number of captured individuals, not the capture index, so there was no need to redo them.

We also added to the Methods section: “To calculate the effective trap-nights we subtracted from the total number of trap-nights those that were non functional and those that were occupied by recaptured individuals.”

Furthermore, to estimate the abundance of species A you need to correct the sampling effort, because traps that captured animals of species B, C, D,….etc., are not available to capture individuals of species A. Missing to do this will generate a bias linked to an overestimation of the trapping effort, that will increase with the number of recaptures and of captures of individuals from distinct species that the one targeted.

Although there is a point in this approach, we do not think it would yield reliable results in this case. For example, in the extreme case when we have 100 TN and 100 individuals captured, among which 99 A. flavicollis and 1 S. araneus, excluding the traps occupied by individuals belonging to the other species would yield 100 individuals/100 TN for both species, which does not reflect their population density.

You mention using a relative occurrence index. Why not use a relative abundance index instead (e.g. Marques, S. F., Rocha, R. G., Mendes, E. S., Fonseca, C., & Ferreira, J. P. (2015). Influence of landscape heterogeneity and meteorological features on small mammal abundance and richness in a coastal wetland system, NW Portugal. European Journal of Wildlife Research, 61(5), 749–761. doi:10.1007/s10344-015-0952-2)? ;

We used the relative occurrence only for the calculation of niche width, where a relative abundance index can not be used, as we need pi – the proportion of the total number of individuals of the considered species that exploit resource i – in our case, which populate habitat type i.

  1. Page 5, Lines 216-217 – What is the publication year of Ter Braak and Smilaeur?

The year is 2018. We corrected the omission.

Again, if the trapping effort differed between patches, how did you compare your results taking into consideration this bias?;

In the CCA, data are automatically standardized by sample (site) total, i.e., results refer not to abundance (number of individuals) but to relative abundance (the proportion of each species within the captured individuals), which is little affected by trapping effort.   

  1. Page 5, Lines 228-232 – All patches were sampled every year? If not, how did you cope with individual variation within patches between years?

We addressed this questions previously, at point 2.

  1. Page 5, Lines 238-244 – I suggest authors to consider the use of Hill numbers, which seem to be a high robust approach for these objectives (Chao, A., Colwell, R. K., Gotelli, N. J., & Thorn, S. (2019). Proportional mixture of two rarefaction/extrapolation curves to forecast biodiversity changes under landscape transformation. Ecology letters, 22(11), 1913-1922. doi:https://doi.org/10.1111/ele.13322).

For q=0 the Hill number is actually the species richness and for q=2 the Hill number is the inverse Simpson index, but we used its compliment, Gini-Simpson, because it is easily calculated using the multivariate ordination methods.

 Discussion

  1. Page 13, Lines 437-438 – What is the ecological explanation for a high density of Crocidura shrews in plantations?;

Actually we can not say that Crocidura shrews had high densities in poplar plantations, but they were present there, in contrast with other types of forests, where they were not captured. The presence of these shrews in poplar plantations results from their habitat characteristics (reduced tree canopy, rich grassy layer), proximity of open habitats and reduced forest patch size and isolation, maybe also their history.

  1. Page 13, Lines 442-443 – Its assumed that plantations support always lower abundances of small mammals than surrounding environments. But this will depend on the type of plantations, implemented management and production cycle. The later if mostly linked to plantation’s temporal heterogeneity in structure (see; Teixeira, D., Carrilho, M., Mexia, T., Köbel, M., Santos, M. J., Santos-Reis, M., & Rosalino, L. M. (2017). Management of Eucalyptus plantations influences small mammal density: Evidence from Southern Europe. Forest Ecology and Management, 385, 25-34. doi:10.1016/j.foreco.2016.11.009; Timo, T. P. C., Lyra-Jorge, M. C., Gheler-Costa, C., & Verdade, L. M. (2014). Effect of the plantation age on the use of Eucalyptus stands by medium to large-sized wild mammals in south-eastern Brazil. iForest - Biogeosciences and Forestry, 8, 108-113. doi:10.3832/ifor1237-008);

Yes, this is true. Our results reflect only the situation in the mature poplar plantations and can not be generalised to other types of plantations.

  1. Page 13, Line 449 – Here authors compare their data with plantations elsewhere. But to what kind of plantations? Sometimes this, low diversity and richness pattern is not the detected pattern. Depends on the landscape context and the plantations’ type, management and structure.

We agree. We added at the end of the paragraph: “However, abundance and composition of small mammal communities depend on the type of plantations, implemented management and production cycle. The later is mostly linked to plantation’s temporal heterogeneity in structure (Timo et al., 2014; Teixeira et al., 2017). Therefore, our results reflect only the situation in mature poplar plantations and can not be generalised to other types of plantations.”

  1. Page 14, Lines 456-460 – Authors infer that the species richness patterns are linked to tree canopy diversity and habitat heterogeneity. But this effect was inferred not tested. I suggest testing this. Using richness as the dependent variable and the stand size, canopy diversity, location of the stand, habitat heterogeneity, as candidate drivers in a GLM or GLMM approach, you can test these inferred hypothesis;

In the revised version of the manuscript, we included two GLMMs with species richness and heterogeneity (Simpson index) as response variables, elevation, diversity of tree canopy and overall habitat heterogeneity as predictors and year as random factor. We included the results (the best models) also in the Discussion section.

  1. Page 14, Lines 482-484 – A more detailed conservation and management implication can be derived from these data. I suggest authors to detail why are you suggesting this and be more clear on what is their advise to future researcher to consider in their monitoring scheme to acquire a more realistic data set.

We changed the paragraph to: “Our results from the forest plot surveyed 60 times show that long term monitoring can yield observed species richness that exceeds richness estimated based on single samplings, even in multiple sites, revealing the real diversity potential of specific habitat types. Therefore, to assess small mammal diversity in temperate forests, communities need to be surveyed each year for a longer period of time (at least 4-5 years).”

  1. Page 14, Line 499 – Vellend 2006 et al in listed in the reference list as Vellend et al. 2007;

The year is 2007. We corrected the error in the citation.

  1. Page 14, Lines 507-511– In this paragraph authors suggest that is important to assess the diversity/richness drivers. I think, as I mentioned earlier, that they can include such approach in the new version of the manuscript.

Yes, we included the analysis of the effect of some potential drivers of diversity, as mentioned before and deleted the mentioned paragraph.